# GLRM: Geometric Layout-Based Resource Management Method on Multiple Field Programmable Gate Array Systems

**Hongxu Gao** [1], **Zeyu Li** [1,2,*] , **Lirong Zhou** [1,*], **Xiang Li** [3] **and Quan Wang** [1]

1   School of Computer Science and Technology, Xidian University, Xi'an 710071, China;
    hx_gao@stu.xidian.edu.cn (H.G.); qwang@xidian.edu.cn (Q.W.)
2   School of Computer Science and Technology, North University of China, Taiyaun 030051, China
3   School of Decision Sciences, The Hang Seng University of Hong Kong, Hong Kong 999077, China;
    p233343@hsu.edu.hk
*   Correspondence: 20230101@nuc.edu.cn (Z.L.); zlrong@stu.xidian.edu.cn (L.Z.);
    Tel.: +86-1500-343-9992 (Z.L.); +86-1882-955-0389 (L.Z.)

**Abstract:** Multiple field programmable gate array (Multi-FPGA) systems are capable of forming larger and more powerful computing units through high-speed interconnections between chips and are beginning to be widely used by various computing service providers. However, the new computing architecture brings new challenges to the system's task resource management. Existing resource management methods do not fully exploit resources in Multi-FPGA systems, and it is difficult to support fast resource request and release. In this regard, we propose a geometric layout-based resource management (GLRM) method for Multi-FPGA systems. First, a geometric layout-based task combination algorithm (TCA) was proposed to ensure that the final system can use the available FPGA resources more efficiently. Then, we optimised two resource management algorithms using TCA. Compared with state-of-the-art resource management methods, TCA increases resource flexibility by an average of 6% and resource utilisation by an average of 7%, and the two optimised resource management methods are effective in improving resource management performance.

**Keywords:** geometric layout; resource management; resource allocation; task combination; Multi-FPGA





## 1. Introduction

Field programmable gate arrays (FPGAs) are gradually replacing X86 or GPU in high performance computing platforms in resource-constrained environments due to their low power consumption, high parallelism, and fast computing speed [1,2]. As applications become larger and more complex, system-on-chip (SoC) architectures consisting of multiple FPGAs (Multi-FPGA) that combine faster inter-chip interconnections to form larger, more computationally intensive units have become popular [3,4]. In addition, the Dynamic Partial Reconfiguration (DPR) technology of FPGA allows the runtime to dynamically configure tasks to different reconfigurable partitions [5], further increasing the flexibility of Multi-FPGA systems and virtually increasing the availability of hardware resources [6,7].

Although the Multi-FPGA architecture and DPR technology provide great flexibility in system design, resource management and the overhead incurred during the reconfiguration process must be carefully considered, as they can easily compromise the performance gains achieved through hardware acceleration [8,9]. The multi-FPGA architecture is like a single FPGA chip, which is a "black box" and invisible to the user, but when implementing applications on it, some issues should be considered: task deployment location, resource constraints and resource reuse, etc. In addition to employing DPR technology, there is a need for further requirements regarding task segmentation and reorganization.

The original FPGA resource management algorithms utilize board-level granularity for management. A typical form of organisation is known as MPC-X [10], which has the advantage of the simplicity of operation and the disadvantage of very low resource

flexibility. To improve the flexibility of resource usage, slot-based and BRAM-based resource management approaches have been proposed [11–13], which reduce the resource granularity compared to FPGA chip-based resource management, with the disadvantage of not being able to use the freed resources. With the development of dynamic partially reconfigurable technology, researchers have proposed centralised bus-based [14] and quadtree-based [15] resource management algorithms with a resource granularity of resource rectangles without fixed boundaries, which can effectively control resource fragmentation, but the flexibility and efficiency of resource utilisation are still low. In addition, with the emergence of Multi-FPGA systems as a new computing architecture, the existing single-chip resource management methods can no longer be adapted to multi-chip FPGA resource management [16–18].

Overall, existing resource management methods suffer from high resource wastage and are not applicable to Multi-FPGA systems. In response, we propose a geometric layout-based resource management method on Multi-FPGA systems, which includes a geometric layout-based task combination algorithm (TCA) and two resource management algorithms optimized by TCA. The proposed method is experimentally verified and demonstrates the effectiveness of resource management. In this context, the contributions can be summarised as follows:

1. A geometric layout-based task combination algorithm to make sure that the final system can make use of the available FPGA resources more efficiently. The available resources and tasks on the FPGA chip are abstracted into resource rectangle models and task rectangle models of specific length and width, and the optimal combination strategy of task rectangle models is generated with the aim of maximising the utilisation of the resource rectangle models.

2. Two resource management algorithms optimized by TCA to improve resource management performance. The improved quadtree resource management algorithm uses variables instead of resource layouts to simulate the allocation process to achieve fast resource request and recovery. The improved central bus resource management algorithm uses the resource fragments generated by task placement to place other tasks to improve resource utilisation.

3. The experimental results show that the task combination algorithm is able to effectively combine and place the tasks, with an average increase of 6% in resource flexibility and an average increase of 7% in resource utilisation; moreover, the two optimised resource management methods can effectively improve the resource management performance.

The paper is structured as follows. Section 2 discusses related work on FPGA resource management. In Section 3, we introduce the geometric layout-based task combination algorithm. In Section 4, we provide a detailed description of the two resource management algorithm optimized by TCA. In Section 5, we evaluate the performance of our method. Section 6 concludes the paper.

## 2. Related Work

The evolution of FPGA resource management can be abstracted as a process of changing resource granularity from large to small. In the early period, FPGA resource management was managed by using board-level granularity, where multiple FPGAs were grouped into a group and the minimum configuration unit (MCU) was a single-chip FPGA [19]. The advantage of the FPGA group is that it is easy to operate and meets the resilience requirements for high-performance computing platforms, but the flexibility of using the on-chip resource is very low [10,20].

With the huge resources of FPGAs and the maturity of DPR technology, the granularity of FPGA resource management gradually tends to parts of the on-chip resources. At this time, the MCU is called "slot", which is a fixed-boundary resource rectangle [11,12]. On this basis, ref. [13] proposed a resource management strategy, the MCU of which is a resource rectangle with no fixed boundary. Due to the lack of fixed shape constraint, the flexibility of resources is promoted, but the management of whole resources is lacking. In addition,

the literature [15] proposed a quadtree algorithm (adaptive reconstruction region), the MCU of which is a resource rectangle with no fixed boundary, and the management mode is shown in Figure 1. This method can effectively control resource fragmentation and ensure connectivity between tasks, but the bus would consume some resources. Another resource management algorithm with no fixed boundary is based on a centralised bus (combined reconstruction region) [14], as shown in Figure 2. This method reduces the resource consumption of the bus, but the one-sided dimension is fixed.

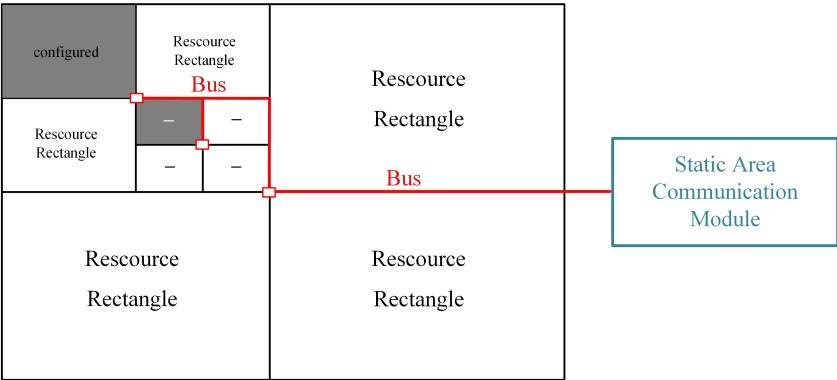

**Figure 1.** Quadtree algorithm.

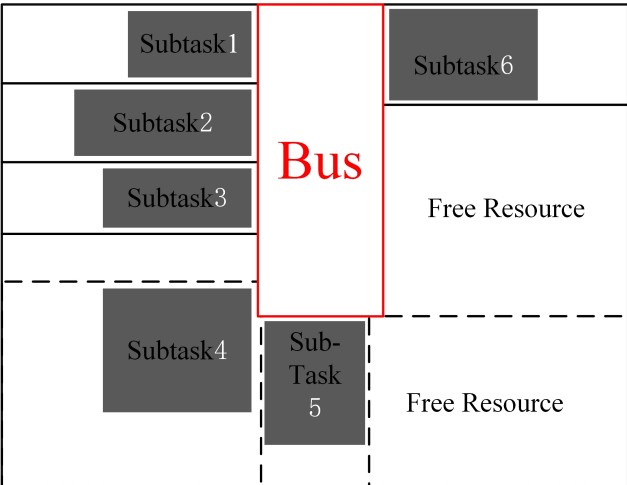

**Figure 2.** Centralized bus algorithm.

In [12], a task encapsulation strategy based on switch is proposed to place some subtasks of one large task on multiple FPGAs, thereby enabling one large task to be placed on multiple FPGAs. However, the configuration file and routing of the networked tasks are complicated, which makes resource management difficult. Ref. [21] proposed a novel area sharing method that leverages the exploratory capabilities of OpenCL, employing intelligent clustering and custom, task-specific partitioning and mapping to more effectively manage the resource requirements of tasks. By selecting the most appropriate distribution that best enhances the temporal computation density based on runtime workload demands, the system's throughput is improved. Mehrabi [22] highlighted the importance of considering spatio-temporal strategies in FPGA resource management to achieve long-term target allocations and optimize FPGA usage efficiency by tracking and correcting deviations.

In this work, we propose a task combination algorithm based on a geometric layout and combine it with two state-of-the-art resource management algorithms. By integrating applications for resources, the resource granularity is further reduced, the resource waste is reduced, and the resource utilization and resource flexibility are improved. In addition, the placement is accelerated by separating the allocation and relocation of resources.

## 3. Geometric Layout-Based Task Combination Algorithm

In this work, we propose a geometric layout-based task combination algorithm that reduces the number of resources occupied by the combination by allowing the tasks to use resource fragments from other tasks in the combination, thus improving the resource efficiency of the FPGA.

### 3.1. Task Combination Algorithm

FPGA resource management is constantly evolving towards more flexibility and less fragmentation. To simplify the resource management process, the task combination problem is abstracted as a rectangle combination problem by abstracting the tasks into a set of rectangles on the FPGA based on their resource requirements. The combined rectangle must contain all the tasks, as well as the buses needed to connect the tasks, to ensure that all the small rectangles can be connected to the system's buses. Suppose there are several small rectangles $S_{mission} = \{Rec_1, Rec_2, Rec_3, \ldots\}$; then, we need to find a set of layouts $S_{mission}\{C_1, C_2, C_3, \ldots\}$, each of which produces a large rectangle containing all the rectangles in which the layout is contained. If any two rectangles are combined, the length of the combining edges is not equal and there will be resource fragments. We set the rectangle set $S_{BIT} = \{Rec_1, Rec_2, \ldots\}$ and use $(W_i, H_i)$ to represent the horizontal and vertical sides of the rectangle. Suppose $W_i > H_i$ and $Waste_{i,j}$ represents the size of the resource fragment generated when $Rec_i$ and $Rec_j$ are combined; then, $Waste_{i,j}$ can be calculated by Formula (1):

$$Waste_{i,j} = \begin{cases} max(H_i, H_j) * (W_i + W_j) - W_i * H_i - W_j * H_j, & Horizontal\,Combination \\ max(W_i, W_j) * (H_i + H_j) - W_i * H_i - W_j * H_j, & Vertical\,Combination \end{cases} \quad (1)$$

According to the existing resource management algorithm, if there are three subtasks $a$, $b$, and $c$, the amount of resources required for subtasks is $R_a$, $R_b$, and $R_c$, respectively. Then, the resources $D_a$, $D_b$, and $D_c$ allocated for each task satisfy $D_a > R_a$, $D_b > R_b$, and $D_c > R_c$, and using the task combination algorithm proposed in this work, the system allocates the resource amount $D_{a+b+c}$ to the combined three subtasks, which satisfies $D_{a+b+c} \geq R_a + R_b + R_c$. The parameter $P$ is used to represent the product of the subtask and the product of the start–end time interval, that is, the weighted resource occupancy of the subtask. As shown in Figure 3a, the start and end times of each subtask in the system are $(S_a, E_a)$, $(S_b, E_b)$, and $(S_c, E_c)$, respectively. The resources occupied in each task are $P_a$, $P_b$, and $P_c$, and the total resources occupied by the three subtasks is shown in Formula (2):

$$P_a + P_b + P_c = D_a * (E_a - S_a) + D_b * (E_b - S_b) + D_c * (E_c - S_c) \quad (2)$$

As shown in Figure 3b, after the combining tasks, the start and end times of the three tasks in the system are all $(S_a, E_b)$, and the total weighted resource occupancy of the task is $P_{a+b+c}$. The calculation method is as follows:

$$P_{a+b+c} = D_{a+b+c} * (max(E_a, E_b, E_c) - min(S_a, S_b, S_c)) \quad (3)$$

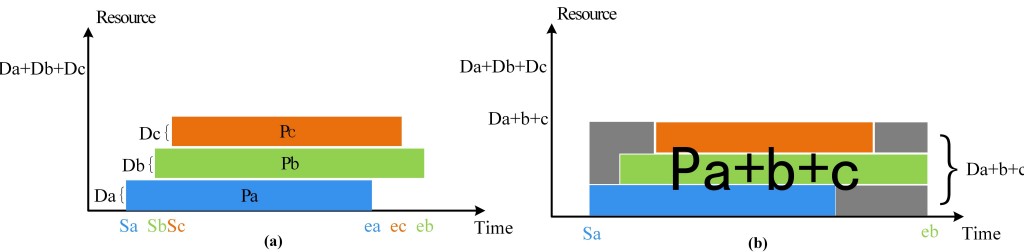

**Figure 3.** Optimization of resource management algorithm performance by task combination. (**a**) Resources occupied of each task before optimization, (**b**) Resource occupation time after task combination.

If the weighted resource occupancy of the resource allocation and the combined allocation meets $P_{a+b+c} < P_a + P_b + P_c$, the number of tasks accommodated by the system at the same time can be increased by the task combination, and the total resource cost of the task is reduced.

*3.2. Implementation of Algorithm*

3.2.1. Build the Sequence of Algorithm

The arrangement $Edge_{BIT} = \{E_1, E_2, E_3, \ldots\}$ of all sides is constructed when the combination of the adjacent sides is performed. If the adjacent edges correspond to different rectangles and are both horizontal or vertical, a new combined rectangle is generated. If the number of elements in $S_{BIT}$ is $n$ and the number of newly generated rectangles is $N_{NEW}$, then $N_{NEW}$ satisfies $N_{NEW} \leq 2n - 1$. A combination of the two edges that satisfy $E_i < E_j$ produces a new rectangle whose side length can be expressed as Formulation (4).

$$Rec_{new} = \begin{cases} (W_i, H_i + H_j), & E_i = W_i \text{ and } E_j = W_j \\ (W_i + W_j, H_j), & E_i = H_i \text{ and } E_j = H_j \end{cases} \tag{4}$$

The maintenance of the edge sequence uses a greedy strategy, and under the premise of including all the rectangles in $S_{BIT}$, a new rectangle with the smallest $Waste_{i,j}$ can be generated.

3.2.2. Two-Dimensional Equilibrium Evaluation

After the rectangular combination operation, several new rectangles are generated, and it is necessary to select the acceptable combinations from these combinations and record the progress of the combination. The characteristics of the min heap satisfy the sorting method required for the rectangular combination selection. In order to compare the advantages and disadvantages of the rectangular combination, a min heap is established by sorting the unacceptable degrees.

We define the ratio of width to height (**RWH**) as the ratio of the lateral side length of the rectangle to the length of the longitudinal side. The formula is expressed as follows:

$$RWH_i = W_i / H_i \tag{5}$$

The standard ratio of width to height (**SRWH**) is the ratio of the horizontal and vertical edges of the resource layout generated by the resource management algorithm. Since the dynamic reconfigurable regions of different FPGA chips have different shapes, the **RWH** of the resource is used as the **SRWH**.

Moreover, combinations of smaller resource fragments and similar shapes and resource shapes should be accepted. We define the acceptance degree (**AD**) as a reasonable degree of the combination that can be expressed as Formulation (6):

$$AD_i = \alpha Waste_i + \beta |RWH_i - SRWH_i| \tag{6}$$

where $\alpha$ denotes the weight of the resource fragment when calculating the acceptability, $\beta$ denotes the weight of the difference value between **RWH** and **SRWH**, and the range of $\alpha$ and $\beta$ is all from 0 to 1.

3.2.3. Maintain the Rectangular Heap Sequence Completion Combination

The details of maintaining the heap structure include the maintenance of the overall progress of the layout, in-heap and out-heap strategies and end conditions, and the maintenance of the edge sequences. The interaction between the heap and the edge sequence, that is, the rectangular combination of the heap into the edge sequence and the new combination of rectangles into the rectangular min heap, covers the entire process of the rectangular combination.

(1) Heap-to-edge sequence operation: The roof element is the current optimal combination scheme and is recorded in the global combined process. The roof element is popped and the heap structure is reconstructed, the four sides of the two sub-rectangles are removed from the edge sequence, and the two sides of the new rectangle are added to the edge sequence.

(2) Edge sequence to heap operation: After the two new edges enter the sequence, respectively, based on the position, a new four-rectangular combination is generated by the new edges and the first edge can be combined, forwards and backwards. Then, the generated new rectangle enters the min heap in turn, and the min heap structure is reconstructed.

Since the rectangles to be combined are not added one by one to the final combination of rectangles, it is likely that the combination will be as shown in Figure 4. Rectangle A is first combined with rectangle B as rectangle E, and rectangle C is combined with rectangle D as rectangle F, and then rectangles E and F are combined. The final layout scheme is composed, so the current layout progress of the min heap record is a variable queue, and each element in the queue represents a segment of the combination process. In addition, once rectangle A has been combined with rectangle B as rectangle E, rectangle A is not accepted for other schemes to generate new rectangles. Only rectangle E and other rectangles are accepted to generate a new rectangle, because rectangle E is the current optimal solution. Furthermore, the combination of rectangle A and other rectangles belongs to the current sub-optimal solution. According to the greedy strategy, only the optimal solution is accepted. The complete procedure of the algorithm is described in Algorithm 1:

---

**Algorithm 1** Combination process based on edge sequences and small top heaps

---

**Input:** Task Rectangle Collection *missions*
**Output:** Combined task shape
 1: **for** *m* in *missions* **do**
 2:      Add the edges of *m* to the collection of edges *edges*
 3: **end for**
 4: Ascending order edges
 5: **for** *e* in *edges* **do**
 6:      **if** *e* with a critical edge if it can be formed into a rectangle **then**
 7:          Rectangle joins two-dimensional equilibrium small top heap *minHeap*
 8:      **end if**
 9: **end for**
10: **while** The top element of the heap does not contain all tasks **do**
11:      **for** *s* in *states* **do**
12:          **while** State of the top element of the heap and s != 0 and state of the top element of the heap and s != s **do**
13:              Element out of heap and rebuild heap structure
14:              **break**
15:          **end while**
16:      **end for**
17:      Accept the top element of the heap, exit the heap, and maintain the heap structure
18:      Update the side sequences
19: **end while**
20: Insertion of buses for task combinations

---

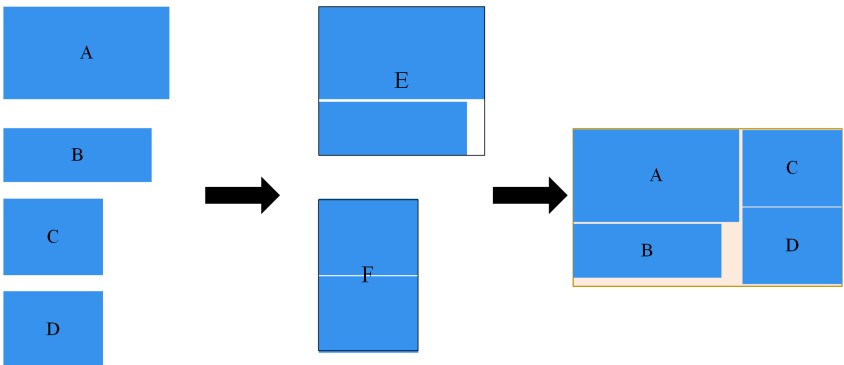

**Figure 4.** The process of rectangular combination.

### 3.2.4. Bus Insert

The insert bus operation is responsible for connecting each task to the global bus. When the rectangle combination is complete, the bus is inserted between the rectangles and all tasks are connected to the system bus.

Assume that no bus rectangle is located in the lower left corner of the blank resource rectangle in the reconfigurable area; then, the bus resource is added from the lower left corner to the upper right corner. As shown in the Figure 5, representing the two regions in coordinates, the space occupied by the busless rectangle is $(0,0) \rightarrow (W_i, H_i)$

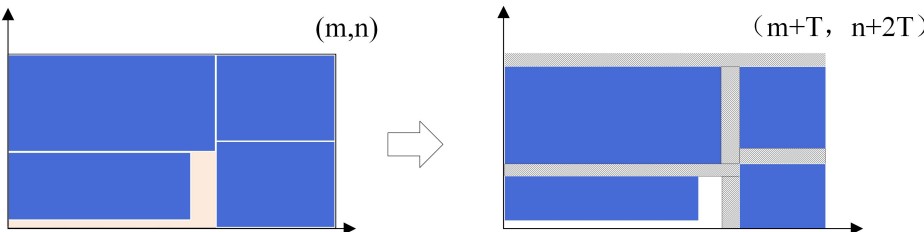

**Figure 5.** Schematic diagram of resource utilization by the bus.

For a rectangle $R$ containing $n$ sub-rectangles, the number of sub-rectangles in the column containing the largest number of sub-barrels in $R$ is $Column_{max}$, and the maximum of the rows is $Row_{max}$. Then, the total height is increased by $Column_{max}$, and the width is increased by $Row_{max}$, so that the increased bus resources can be connected to all child rectangles with certainty. For a rectangle $R$ containing $n$ sub-rectangles, if the total height increase is less than $\lceil Column_{max}/2 \rceil$ or the total width increases by less than $\lceil Row_{max}/2 \rceil$, then it is not guaranteed that all rectangles are connected to the bus; meanwhile, when the total height is increased by $\lceil Column_{max}/2 \rceil$ and the total width is increased by $Row_{max}$, or when the total width is increased by $\lceil Row_{max}/2 \rceil$ and the total height is increased by $Column_{max}$, it can be ensured that the increased space can place a bus that can be connected to all rectangles.

As shown in Figure 6, we can choose to apply the bus resources along the positive direction of the x-axis or the positive direction of the y-axis. Select the corresponding resource application method according to the number of blank resources in both directions.

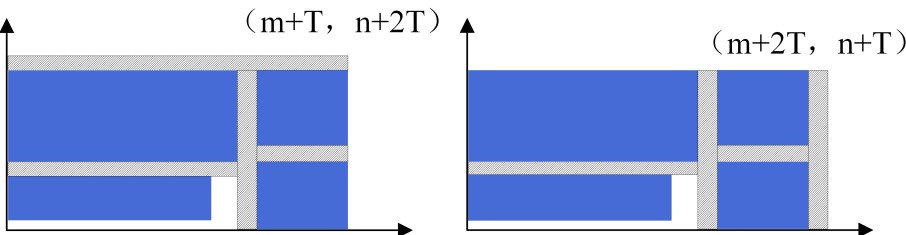

**Figure 6.** Horizontal application and vertical application.

The total amount of resources applied is expressed by the following Formula (7).

$$(W_{req}, H_{req}) = \begin{cases} (W + Row_{max}, H + \lceil Column_{max}/2 \rceil), & W/H < W_{req}/H_{req} \\ (W + \lceil Row_{max}/2 \rceil, H + Column_{max}), & else \end{cases} \quad (7)$$

As shown in Algorithm 2, the process of inserting the bus is described as follows:

---

**Algorithm 2** The process of inserting the bus

---

**Input:** Collection of tasks, shape of resources
**Output:** Combination of tasks after insertion of bus
  1: Combination according to uninserted and inserted buses
  2: Calculate the shape difference between the two combinations to obtain the maximum number of row and column rectangles
  3: **if** Follow the width and height of the no-bus to find the right resource **then return** false
  4: **end if**
  5: **if** More vertical white space in the selected resource area **then**
  6:     Requests for resources under the vertical application modality
  7: **else**
  8:     Request for resources under the horizontal application modality
  9: **end if**
 10: **if** Failed application **then return** false
 11: **end if**

---

*3.3. Complexity Analysis*

The algorithm strictly adheres to the optimal substructure and reduces the number of invalid combinations. Each combination will reduce the possible combination, with $N$ indicating the number of tasks to be combined, the time complexity of the combined algorithm is $O(logN)$, the time complexity of maintaining the heap structure is $O(logN)$, and the final algorithm time complexity is $O(logN^2)$. The algorithm needs to create heap space for all possible combinations that occur during the combination process, with a space complexity of $O(N^2)$.

**4. The Quadtree and Central Bus Resource Management Algorithm Optimized by TCA**

*4.1. Improved Quadtree Resource Management Algorithm*

The traditional tree resource management structure has low efficiency due to the overhead of maintaining the tree structure and the resource waste caused by different aspect ratios of the task rectangle. To reduce the effect of rectangular aspect ratio, tasks are combined using task combination algorithms. This paper proposes an improved method of quadtree resource management that uses the count method instead of the tree structure to reduce the overhead of maintaining the tree structure. Using array-based containers instead of pointers can improve memory management efficiency and reduce program space and time complexity.

(1) CLB resource constraints: the CLB resources applied by different subtasks cannot overlap at the same time to prevent mutual interference. To satisfy the CLB resource constraint, the horizontal spacing of the starting coordinates of the two rectangles is greater than the width or longitudinal spacing of one of the rectangles, which is greater than the height of one of the rectangles. The formula is expressed as follows:

$$\begin{cases} |x_1 - x_2| < w_1 \,\|\, |x_1 - x_2| < w_2 \\ |y_1 - y_2| < h_1 \,\|\, |y_1 - y_2| < h_2 \end{cases} \quad (8)$$

where $|x_1 - x_2|$ represents the horizontal spacing between the starting coordinates of the two rectangles, $w_i$ represents the width of rectangle $i$, and $h_i$ represents the height of rectangle $i$.

(2) Bandwidth constraints: bandwidth constraints are divided into bus bandwidth constraints and on-chip resource bandwidth constraints. The division of the bus determines the communication time, further affecting the total length of execution time of the subtask. The formula is expressed as follows:

$$B_1 - B_2 > Trans_1 \tag{9}$$

(3) Storage constraints: the storage constraints need to consider the spacing of storage resources. For a subtask that requires on-chip storage resources, the spacing determines the minimum resource level that it can request, that is, the resource level contains at least one storage resource.

By taking the logarithm of the total resources with a base of 4, resource requests are graded, which is more in line with the quadtree resource management approach and also convenient for computation.The "level-resource amount" table is created with the level as the subscript, and the number of resources contained in the node of the current level is stored. As shown in Figure 7, the entire reconfigurable partition is treated as a level 0 resource, and the resource level is incremented by one each time. When the quadtree is in its initial state, the number of level 0 resources is 1, and the number of other equivalent resources is 0. Instead of maintaining the quadtree structure, use the resource node amount maintenance method. A container whose size is the total number of levels is a "level-node amount" table; the table entry stores the amount of resources corresponding to the current resource level and the number of remaining nodes. When a subtask requests a resource, it can obtain a level suitable for its own needs and request resources by checking the level-node amount table.

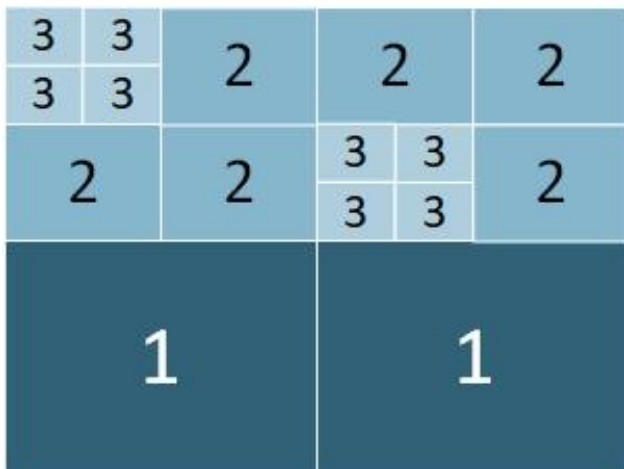

**Figure 7.** Resource level schematic.

The specific location of the resource is determined based on the task sequence. When the resource application process is saved by the vector, if there are four resource rectangles in a particular level, the resource rectangles are merged. When a quadtree is moved, a specific resource area must be allocated to the task in the order in which the resources are released.

A quadtree is generated according to the resource level and the occupied time period corresponding to the subtask rectangle. Sort each subtask rectangle by release time, which refers to the time when the task requests resources. For subtasks with the same release time, resources are allocated to subtasks with earlier application resources, and the quadtree is generated according to depth at first search. The complete procedure of the algorithm is described in Algorithm 3.

---

**Algorithm 3** Delayed generation of quadtrees

---

**Input:** Interval during which the task is inside the device
**Output:** Generate a quadtree
  1: Create a rectangular sequence of tasks for which resources are to be requested
  2: Sort the task rectangles linearly in the order of when the tasks release their resources
  3: Sorted in order of when resources were requested
  4: Stable sorting by release time
  5: Sort the task rectangle by the time heap of the task requesting resources
  6: **while** All rectangles in the rectangle sequence are not all traversed **do**
  7:     **while** Resource release time > time to request the top element of the heap **do**
  8:         Release resources based on the level of resources requested by the mandate
  9:         Remove nodes from the task heap for which resources have been requested
 10:         Select subtrees in order of priority from smallest to largest numbering
 11:         Add tasks to the heap of tasks for which resources have been requested, maintaining the heap structure
 12:     **end while**
 13: **end while**

---

### 4.2. Improved Centralized Bus Resource Management Algorithm

In this paper, we further propose an optimization method for the centralized bus resource management algorithm based on the task layout algorithm, which uses resource fragments generated by task placement to place other tasks to improve resource utilization. As shown in the Figure 8, the resources are shared after the tasks are combined.

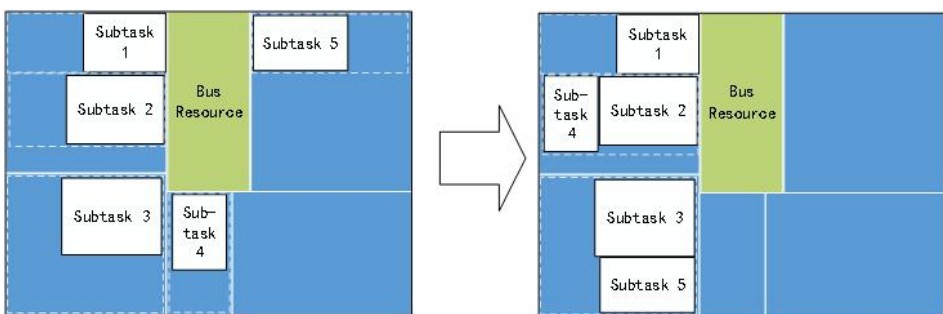

**Figure 8.** Task combination under centralized bus.

Unlike the improved quadtree algorithm, which uses the *RWH* of the reconfigurable area as the *SRWH*, the centralized bus resource management algorithm has no fixed *SRWH*, and the *SRWH* changes as the shape of the resource changes. Therefore, the improved algorithm accepts a combination of side lengths as close as possible to the resource to determine the side length, regardless of *SRWH*. The degree of non-acceptance can be expressed as follows:

$$AD_i = \alpha Waste_i + \frac{\beta}{RWH} \tag{10}$$

where $AD_i$ is the unacceptance degree of rectangle $i$, $\alpha$ denotes the weight of the resource fragment when calculating the acceptability, and $\beta$ denotes the weight of the difference value between RWH and SRWH.

There are different settings for the aspect ratio of the combination to adjust the tendency of the shape of the combined task to evolve. The limiting edge length makes the combined shape change towards the result of approaching a certain length as soon as possible to reduce the generation of resource fragments. If a combination makes the resource less efficient, the design is not accepted.

Since the resource size is not strictly granular, the resource management process cannot be simplified by simplifying the resource description. Therefore, the improved centralised bus resource management algorithm does not adopt the design of separating the allocated

resources and determining the resource location. As a result, the time complexity of the centralised bus resource management algorithm for the resource application and release process is higher than that of the quadtree resource management algorithm.

## 5. Experimental Results and Analysis

In this section, we use the Task Graphs for Free (TGFF) to generate task DAG graphs. Then, based on the resource requirements of commonly used algorithms, the effectiveness of the improved resource management algorithm in improving performance was verified.

### 5.1. Testing Environment

We use C++ to build an experimental platform to provide the system framework and function preset interface for mentioned algorithms of task sorting, multiple sequence maintenance, resource management, task combination, and simulated annealing, based on which the algorithm proposed in this work and the comparison algorithm are implemented. The test task set and device network structure adopt three task topology diagrams and two physical FPGA chips to simulate the problem scenario of multi-task scheduling on Multi-FPGA, and the algorithm program is designed in C++ language.

We set the common algorithms for target recognition and image processing as task topology generated by TGFF [23]. The resource requirements of the tasks are shown in Table 1 [24]. In Table 1, common algorithms for target recognition include Debayer, Rectifier, Stereo match, Disparity, Flex-SURF, and Motor Control. Common algorithms for image processing include FPN correction, Dark field corr., FFT, Bad pixel/spike CCSDS 122, Binning, Hough Transform, and Median Filter. Slice represents the unit of measure for logic cells in FPGAs. The full name of BRAM is block RAM. BRAM is the predefined hardware resource in the FPGA that is dedicated to storage.

**Table 1.** List of common algorithm resource requirements.

|  | Algorithm | Slices | BRAM |
|---|---|---|---|
| Target Recognition | Debayer (2x) | 200 | 2 |
|  | Rectifier (2x) | 500 | 30 |
|  | Stereo match | 2500 | 30 |
|  | Disparity | 1000 | 15 |
|  | Flex-SURF | 1000 | 0 |
|  | Motor Control (3x) | 200 | 0 |
| Image Processing | FPN correction | 100 | 0 |
|  | Dark field corr. | 200 | 1 |
|  | FFT | 800 | 7 |
|  | Bad pixel/spike | 100 | 2 |
|  | CCSDS 122 | 2500 | 12 |
|  | Binning | 300 | 4 |
|  | Hough Transform | 1800 | 14 |
|  | Median Filter | 800 | 0 |

The width and height of the task are generated by random numbers based on the amount of logical resources in order to complete the simulation of the real shape of tasks. Considering that, the resource management algorithm studied in this work may be partially limited by the underlying layout and routing rules. Therefore, we validated the experiment in the reconfigurable virtual layer without considering the rule limitations of specific FPGA chips.

### 5.2. Evaluation Metrics

In order to evaluate the performance of resource management algorithms, three evaluation metrics are introduced below:

(1) Scheduling Flexibility (SF): Indicates the amount of tasks placed at the same time as a percentage of the total amount of tasks; the expression is as follows.

$$SF = \frac{1}{s} \sum_{i=1}^{s} \frac{M_i}{M} \tag{11}$$

where $s$ represents the amount of sequences, $N$ represents the maximum amount of tasks on the chip, and $M$ represents the total amount of tasks.

(2) Resource Efficiency (RE): Indicates the rationality of the task $M$ deployed in the resource area $R$. It is used to evaluate the resource utilization of different FPGA on-chip resource management algorithms. The expression is as follows.

$$RE = \sum_{i=1}^{p} \left( \frac{R_i}{\sum_{j=1}^{p} R_j} * \frac{M_i}{R_i} \right) = \frac{\sum_{i=1}^{p} M_i}{\sum_{j=1}^{p} R_j} \tag{12}$$

where $p$ is the amount of tasks, $M$ is the amount of resources requested by the task, and $R$ is the amount of resources acquired by the task.

(3) Biggest Resource Area (BRA): It represents the largest available free resource rectangle on the FPGA chip, i.e., the maximum resource request that the FPGA chip can respond to. For quadtree resource management, the BRA is shown in area A in Figure 9a. For centralized bus resource management, the BRA is shown in area C in Figure 9b.

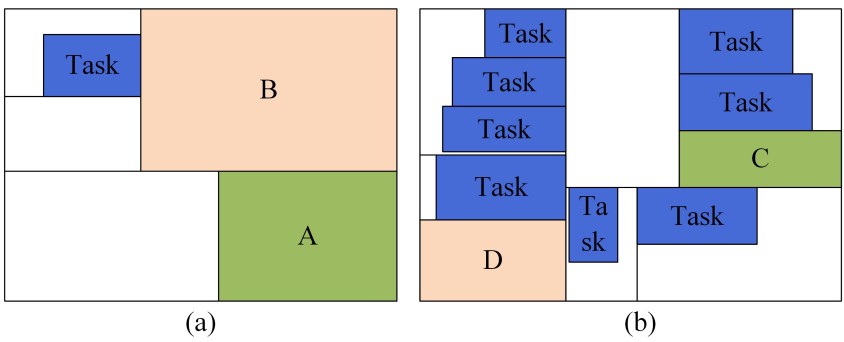

(a)                    (b)

**Figure 9.** The BRA under the two resource management algorithms. (**a**) BRA for quadtree resource management, (**b**) BRA for centralized bus resource management.

### 5.3. Results and Discussion

We simulate the allocation of resources according to the task sequence, which is the sequence generated by the full alignment of the 11 tasks, and then generate resource layouts according to the resource management algorithms before and after improvement, respectively. Before simulating the allocation of resources, the device is considered empty, and while simulating the allocation of resources, it becomes non-empty. As mentioned above, the two improved algorithms have different advantages in managing resources in empty and non-empty devices. Therefore, in addition to comparing the performance indices of the two sets of original and improved algorithms, the design experiment also compares the performance of the four algorithms in the two resource scenarios. In the empty device, the comparison of the performance indices of the four resource management algorithms is shown in Table 2.

First, we compare the original algorithm with the improved algorithm. In response to the improvement of the quadtree resource management algorithm, the resource utilisation rate has increased by 3.6%, the resource utilisation rate of the centralised bus resource management algorithms has improved by 10.2%, and the performance improvement of the centralised bus algorithm is more obvious. The improved quadtree algorithm has a 6.2% increase in scheduling flexibility, and the improved centralised bus algorithm has an increase of 100%. Due to the increased resource utilisation, the improved algorithm can

leave more space for more tasks. When resources are sufficient, the BRA is the largest task that the system can accommodate, and the data are the averages generated by different sequences. As can be seen from the table, the improved algorithm significantly increases the size of the free area.

**Table 2.** Experimental results of four resource management algorithms on a blank device.

| Resource Management Algorithm | Resource Utilization Rate | Scheduling Flexibility | Largest Available Free Resource |
|---|---|---|---|
| Quadtree Resource Management algorithm [15] | 26.8% | 60.4% | 1256 |
| Improved Quadtree Resource Management Algorithm | 30.4% | 66.6% | 3447 |
| Centralized Bus Resource Management Algorithm [14] | 46.4% | 97.1% | 5281 |
| Improved Centralized Bus Resource Management Algorithm | 56.6% | 100% | 5821 |

Second, we compare the two sets of algorithms. Although the improved quadtree algorithm has improved the indicator data, there is still a significant gap with the original centralised bus algorithm. According to the analysis, the centralised bus algorithm has obvious advantages in managing empty resources to complete the current space. However, if fragmentation occurs in the system, its performance will decrease.

For devices that have been working for some time, some discontinuous resources are already present in the device. In this case, the ability of the four algorithms to tolerate release is compared. The performance comparison of the four algorithms is shown in Table 3 and Figure 10.

**Table 3.** Experimental results of four resource management algorithms on non-blank devices.

| Resource Management Algorithm | Average Resource Utilization Rate | Average Scheduling Flexibility |
|---|---|---|
| Quadtree Resource Management Algorithm [15] | 26.9% | 46.0% |
| Improved Quadtree Resource Management Algorithm | 37.2% | 75.6% |
| Centralized Bus Resource Management Algorithm [14] | 45.3% | 65.0% |
| Improved Centralized Bus Resource Management Algorithm | 55.0% | 96.3% |

The changes in the scheduling flexibility of the application release sequence are shown in Figure 10. According to the results of the second set of experiments, it can be seen that the random application and release of resources changes the numerical relationship between the centralised bus algorithm and the quadtree algorithm. Figure 10 shows the reusability of the generation layout by the quadtree resource management algorithm. Since the task combination algorithm can change the rectangular shape of the application resource to adapt to the resource layout at different times, there is greater flexibility in managing non-empty devices than the original resource management algorithm, as the resource flexibility of the two improved algorithms is always higher than the two original algorithms. The quadtree algorithm is a resource management algorithm with fixed resource shape and size. Multiple tasks in the task sequence share the same task space, and the freed resources are more easily used by subsequent tasks, as shown in Figure 10 by the decreasing gap between the two quadtree algorithms and the original centralised bus algorithm.

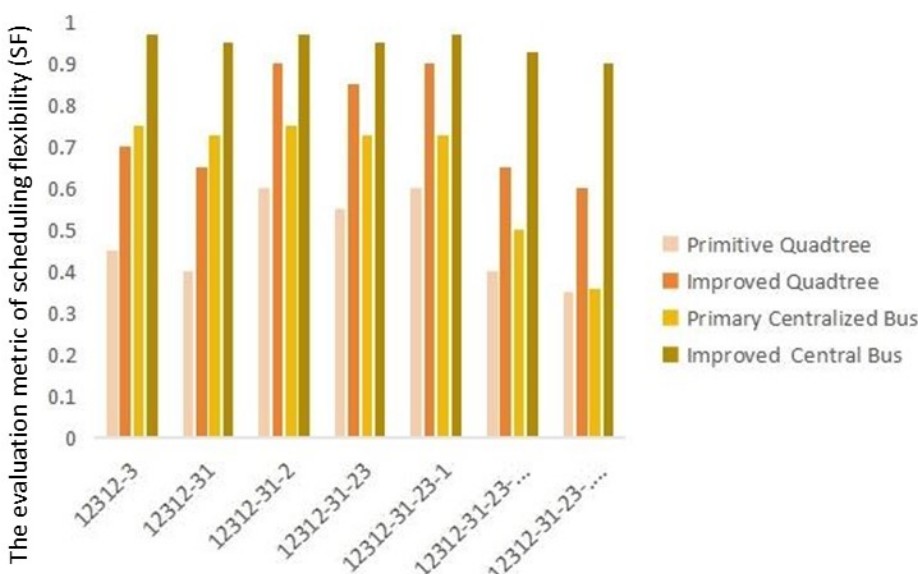

**Figure 10.** The change in scheduling flexibility for a set of release sequences.

### 6. Conclusions

In response to the issue of low resource utilization in multi-FPGA systems, we propose a geometric layout-based resource management method to improve the resource management efficiency of the system. By comparing the differences in performance indicators such as resource utilization, resource flexibility, and maximum available blank area among improved algorithms, quadtree-based resource management algorithms, and centralized bus-based resource management algorithms, the results showed that dynamic task combination can effectively combine and place tasks, with an average increase in resource flexibility of 6% and an average increase in resource utilization of 7%. In future work, we will focus on task placement methods under resource-constrained conditions to introduce appropriate constraints to ensure the effectiveness of resource management methods.

**Author Contributions:** Conceptualization, Q.W.; methodology, H.G. and Z.L.; software, H.G. and L.Z.; validation, Z.L. and X.L.; formal analysis, X.L.; investigation, H.G. and L.Z.; resources, Z.L. and H.G.; data curation, L.Z. and H.G.; writing—original draft preparation, H.G. and L.Z.; writing—review and editing, Z.L.; visualization, X.L.; supervision, Q.W.; project administration, Q.W.; funding acquisition, Q.W. All authors have read and agreed to the published version of the manuscript.

**Funding:** This work was supported in part by the National Natural Science Foundation of China under Grant 61972302, in part by the Guangzhou Municipal Science and Technology Project under Grant SL2022A04J00404, in part by the Fundamental Research Funds for the Central Universities under Grant XJS220306, in part by the Natural Science Basic Research Program of Shaanxi under Grant 2022JQ-680, and in part by the Key Laboratory of Smart Human Computer Interaction and Wearable Technology of Shaanxi Province.

**Data Availability Statement:** All data is contained within the article.

**Conflicts of Interest:** The authors declare no conflict of interest.

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
