# Peer review of "GLRM: Geometric Layout-Based Resource Management Method on Multiple Field Programmable Gate Array Systems"

_electronics, doi:10.3390/electronics13101821_

Round 1
Reviewer 1 Report
Comments and Suggestions for Authors
The proposal is novel, however I think that some sections should be improved for greater clarity of the present work.
The introduction and related work sections can be improved by incorporating more references for example (https://research-information.bris.ac.uk/en/studentTheses/exploring-hardware-support-for-resource-management-in-the-data -EC).
Section 3.2 should be revised, correcting some errors in the mathematical expressions (e.g. lines 147-148) and explaining how the alpha and beta weights are established (168, 169). I think that in section (3) Bus insert, it is possible to explain more clearly and graphically what is described in lines 206 to 216.
On the other hand, it is always important to introduce new variables or terms in the expressions included in the document, for example eq. (5)-(7).
The experimentation section is unclear. In lines 305 -307 it is commented three topologies and the use of FPGAs to simulate the problem scenario, however it is not specified. Some tasks are presented in Table 1, however, it is unclear if they are used for experimentation and how it is done. It is also commented that a specific FPGA is not used. The comments on lines 313-315 are not clear. It is important to clearly establish the experimental mechanism that allows to reproduce the results.
The mechanism to simulate the allocation of resources, the sequence of tasks, and the generation of resource layouts remains unclear. It would be important to describe the mechanism used.
It is also commented that the width and height of the task resources are generated randomly (312-313). What type of generator was used and what parameters were considered? Can be presented the generated values of the task resources?
The conclusions can be improved by emphasizing the results obtained.
Author Response
To Reviewer 1:
The authors appreciate very much for your constructive comments on our manuscript, “GLRM: Geometric Layout-based Resource Management Method on Multi-FPGA Systems” (Manuscript ID Electronics-2959858).
We have revised our manuscript according to your comments. The revision details are summarized as follows.
--------------------------------------------------------
Reviewer #1, Concern #1: The introduction and related work sections can be improved by incorporating more references for example (https://research-information.bris.ac.uk
/en/studentTheses/exploring-hardware-support-for-resource-management-in-the-data -EC).
Author response: Thank you very much for your constructive comment. We conducted a more extensive search on the suggested website and other academic websites to further supplement for relevant literature on the topic of multi FPGA resource management discussed in the paper. However, limited by the niche of research content, there is only a small amount of relevant research progress available for reference and comparison. For this, we have added a few new relevant literature and improved the introduction and related work sections.
Author action: To carefully address your comment, the additions are in lines 99-106.
[21] proposed a novel area sharing method that leverages the exploratory capabilities of OpenCL, employing intelligent clustering and custom, task-specific partitioning and mapping to more effectively manage the resource requirements of tasks. By selecting the most appropriate distribution that best enhances the temporal compute density based on runtime workload demands, the system’s throughput is improved. Mehrabi[22] highlighted the importance of considering spatio-temporal strategies in FPGA resource management to achieve long-term target allocations and optimize FPGA usage efficiency by tracking and correcting deviations.
Reviewer #1, Concern #2: Section 3.2 should be revised, correcting some errors in the mathematical expressions (e.g. lines 147-148) and explaining how the alpha and beta weights are established (168, 169). I think that in section (3) Bus insert, it is possible to explain more clearly and graphically what is described in lines 206 to 216.
Author response: Thank you very much for your constructive comment. We have checked and corrected the mathematical expressions in the paper. α represents the weight of resource fragments in calculating acceptability, β represents the weight of the difference between RWH and SRWH, the range of values is all from 0 to 1.
Author action: To carefully address your comment, we have added the algorithm3.2 to Section3.2.4。
Reviewer #1, Concern #3: On the other hand, it is always important to introduce new variables or terms in the expressions included in the document, for example eq. (5)-(7).
Author response: Thank you very much for your constructive comment. we have made modifications and additions to the expressions and variable explanations of these three formulas.
Author action: To carefully address your comment, we have made modifications in eq.(8)-(10).
Reviewer #1, Concern #4: The experimentation section is unclear. In lines 305 -307 it is commented three topologies and the use of FPGAs to simulate the problem scenario, however it is not specified. Some tasks are presented in Table 1, however, it is unclear if they are used for experimentation and how it is done. It is also commented that a specific FPGA is not used. The comments on lines 313-315 are not clear. It is important to clearly establish the experimental mechanism that allows to reproduce the results. It is also commented that the width and height of the task resources are generated randomly (312-313). What type of generator was used and what parameters were considered? Can be presented the generated values of the task resources?
Author response: Thank you very much for your constructive comment. The attributes of some tasks in the experiment are listed in Table 1, where Slices and BRAM represent the resource requirements of the tasks. Slices and BRAM jointly determine the length and width of task resource representation. It is worth noting that in order to ensure the validity of the experimental results, we randomly selected from numerous tasks, represented the dependency relationships of the tasks based on the DAG graph generated by TGFF, and placed the randomly selected task resources using the method described in this paper. Considering that the resource management algorithm studied in this work may be partially limited by the underlying layout and routing rules. Therefore, we validated the experiment in the reconfigurable virtual layer without considering the rule limitations of specific FPGA chips.
Author action: To carefully address your comment, we have supplemented and modified the issues mentioned in the relevant experimental section on lines 335-340.
“Considering that the resource management algorithm studied in this work may be partially limited by the underlying layout and routing rules. Therefore, we validated the experiment in the reconfigurable virtual layer without considering the rule limitations of specific FPGA chips.”
Reviewer #1, Concern #5: The mechanism to simulate the allocation of resources, the sequence of tasks, and the generation of resource layouts remains unclear. It would be important to describe the mechanism used.
Author response: Thank you very much for your constructive comment. We have explained and supplemented the mechanisms for simulating resource allocation, task sequence, and resource layout generation.
Author action: To carefully address your comment, we have added the clear description to Section5.3.
“We simulate the allocation of resources according to the task sequence which is the sequence generated by the full alignment of the 11 tasks and then generate resource layouts according to the resource management algorithms before and after improvement respectively.”
Reviewer #1, Concern #6: The conclusions can be improved by emphasizing the results obtained.
Author response: Thank you very much for your constructive comment. We have revised the conclusion section and supplemented the focus of future work.
Author action: To carefully address your comment, we have supplemented the relevant experiments in the paper.
“In response to the issue of low resource utilization in multi-FPGA systems, we propose a geometric layout-based resource management method to improve the resource management efficiency of the system. By comparing the differences in performance indicators such as resource utilization, resource flexibility, and maximum available blank area among improved algorithms, quadtree based resource management algorithms, and centralized bus based resource management algorithms, the results showed that dynamic task combination can effectively combine and place tasks, with an average increase in resource flexibility of 6% and an average increase in resource utilization of 7%. In future work, we will focus on task placement methods under resource constrained conditions to introduce appropriate constraints to ensure the effectiveness of resource management methods.”
Reviewer 2 Report
Comments and Suggestions for Authors
The manuscript proposes a geometric layout-based resource management (GLRM) method on Multi-FPGA systems. The results are compared with the state-of-the-art resource management methods, increasing the resource flexibility by an average of 6 % and the resource utilization by an average of 7%.
The manuscript clearly outlines its goals, the results are consistent with whats is proposed and the English is comprehensible. Nevertheless, the article's organization is rather confusing, and it would benefit from a more structured introduction and additional experiments. Detailed considerations regarding the text are provided below
Abstract:
The abstract is very clear and effectively outlines the objectives and potential comparisons with the state-of-the-art. However, it is important to emphasize the names of the two proposed algorithms, namely: the Improved Quadtree Resource Management Algorithm and the Improved Centralized Bus Resource Management Algorithm.
1-) Introduction:
a-) The introduction initially provides a thorough description of the advantages of a Multi-FPGA architecture and DPR technology, as well as their limitations. However, at line 33, the author begins to discuss the limitations of the original FPGA system, which deviates from the manuscript's focus by blending the limitations of Multi-FPGA with FPGA systems. The only statement related to Multi-FPGA limitations in describe from line 47 to 50, "Overall, existing resource management methods suffer from high resource wastage and are not applicable to Multi-FPGA systems. In response, we propose a geometric layout-based resource management method on Multi-FPGA systems, which includes geometric layout-based task combination algorithm (TCA) and two resource management algorithms optimized by TCA." Upon initial review, the state-of-the-art section of the manuscript should be revised to concentrate on similar works specifically addressing only to Multi-FPGA Systems.
b) Key metrics such as RWH, SRWM, and UAD should already be discussed in the introduction and correlated with existing literature, as they represent critical components of the proposed algorithm.
2-) Related Work:
I find this section unnecessary as it duplicates elements already covered in the introduction. For instance, there is no need to reiterate the limitations of FPGA, as stated from line 76 to 87. If the authors wish to emphasize certain restrictive aspects of resource management, it should be integrated into the subsequent sections within the context of the work.
3-) Implementation of Algorithm
3.2.1 - Build the sequence of Algorithm
(a) There is a missing character in the end of line 149. Also the sentence "A combination of the two edges that satisfy Ei < Ej produces a new rectangle whose side length can be expressed as formulation" should be revised.
(b) Since this is a mathematical section, the symbols should be reviewed. 'N_New' and 'SBit' are written differently and should be standardized.
4-) Results
(a) The results section is very well described, showcasing the advantages of both the Improved Quadtree Resource Management Algorithm and the Improved Centralized Bus Resource Management Algorithm compared to their non-improved counterparts. The week point are the limited number of experiments, that should be improved.
(b) The description of the algorithm in Figure 10, which utilizes the term "primary," should match that in Table 2 to avoid confusion. Please standardize accordingly.
(c) Please provide an explanation in the text regarding the x-axis of Figure 10 and the methodology behind its selection. This should be related with the experiments proposed in Table 2.
(d) Considering that the results center around an algorithm, I recommend that the author includes the algorithm as supplementary material. This would enable readers to replicate the obtained results. Additionally, it would enhance the citation of the work in the long term.
Comments on the Quality of English LanguageOn line 31 - "Together with the application of DPR technology,further requirements for task segmentation and reorganisation are proposed.". Make the phrase clearer. I believe the word 'needed' would be more appropriate. Sugestion: "In addition to employing DPR technology, there is a need for further requirements regarding task segmentation and reorganization"
On line 33 - "The original FPGA resource management algorithms used board-level granularity for management." I believe that in this context, using the word "utilize" in the present is more appropriate than using "used" in the past.
On line 334 - "the device is an empty device, and during analogue allocation it is a non-empty device." should be replaced by "the device is considered empty, and during analog allocation, it becomes non-empty.
Author Response
============== Authors’ responses to the reviewer #2’ comments ==============
To Reviewer 2:
The authors appreciate very much for your constructive comments on our manuscript, “GLRM: Geometric Layout-based Resource Management Method on Multi-FPGA Systems” (Manuscript ID Electronics-2959858).
We have revised our manuscript according to your comments. The revision details are summarized as follows.
--------------------------------------------------------
Reviewer #2, Concern # 1: The introduction initially provides a thorough description of the advantages of a Multi-FPGA architecture and DPR technology, as well as their limitations. However, at line 33, the author begins to discuss the limitations of the original FPGA system, which deviates from the manuscript's focus by blending the limitations of Multi-FPGA with FPGA systems. The only statement related to Multi-FPGA limitations in describe from line 47 to 50, "Overall, existing resource management methods suffer from high resource wastage and are not applicable to Multi-FPGA systems. In response, we propose a geometric layout-based resource management method on Multi-FPGA systems, which includes geometric layout-based task combination algorithm (TCA) and two resource management algorithms optimized by TCA." Upon initial review, the state-of-the-art section of the manuscript should be revised to concentrate on similar works specifically addressing only to Multi-FPGA Systems.
Author response: Thank you very much for your constructive comment. We conducted a more extensive search to further supplement for relevant literature on the topic of multi FPGA limitations and resource management discussed in the paper. However, limited by the niche of research content, there is only a small amount of relevant research progress available for reference and comparison. For this, we have added a few new relevant literature and improved the introduction and related work sections.
Author action: To carefully address your comment, the additions are in lines 99-106.
“[21] proposed a novel area sharing method that leverages the exploratory capabilities of OpenCL, employing intelligent clustering and custom, task-specific partitioning and mapping to more effectively manage the resource requirements of tasks. By selecting the most appropriate distribution that best enhances the temporal compute density based on runtime workload demands, the system’s throughput is improved. Mehrabi[22] highlighted the importance of considering spatio-temporal strategies in FPGA resource management to achieve long-term target allocations and optimize FPGA usage efficiency by tracking and correcting deviations.”
Reviewer #2, Concern # 2: Key metrics such as RWH, SRWM, and UAD should already be discussed in the introduction and correlated with existing literature, as they represent critical components of the proposed algorithm.
Author response: Thank you very much for your constructive comment.
Author action: To carefully address your comment.
Reviewer #2, Concern # 3: I find this section unnecessary as it duplicates elements already covered in the introduction. For instance, there is no need to reiterate the limitations of FPGA, as stated from line 76 to 87. If the authors wish to emphasize certain restrictive aspects of resource management, it should be integrated into the subsequent sections within the context of the work.
Author response: Thank you very much for your constructive comment. We have reorganized the section of related work and expanded the sixth section to include descriptions of future work.
Author action: To carefully address your comment, we have made modifications in Section2.
Reviewer #2, Concern # 4: There is a missing character in the end of line 149. Also the sentence "A combination of the two edges that satisfy Ei < Ej produces a new rectangle whose side length can be expressed as formulation" should be revised.
Author response: Thank you very much for your helpful comment. We have supplemented the characters and modified the relevant statements.
Author action: To carefully address your comment, we have made modifications to this sentence in lines(156-157).
“A combination of the two edges that satisfy E_i < E_j produces a new rectangle whose side length can be expressed as formulation(4).”
Reviewer #2, Concern # 5: Since this is a mathematical section, the symbols should be reviewed. 'N_New' and 'SBit' are written differently and should be standardized.
Author response: Thank you very much for your helpful comment. We have checked the symbols in the paper and modified their format.
Author action: To carefully address your comment, we have made modifications to the relevant symbols in line 155.
“If the number of elements in S_BIT is n, and the number of newly generated rectangles is N_NEW , then N_NEW satisfies NNEW ≤ 2n − 1.”
Reviewer #2, Concern # 6: (a) The results section is very well described, show casing the advantages of both the Improved Quadtree Resource Management Algorithm and the Improved Centralized Bus Resource Management Algorithm compared to their non-improved counterparts. The week point are the limited number of experiments, that should be improved. (b) The description of the algorithm in Figure 10, which utilizes the term "primary," should match that in Table 2 to avoid confusion. Please standardize accordingly. (c) Please provide an explanation in the text regarding the x-axis of Figure 10 and the methodology behind its selection. This should be related with the experiments proposed in Table 2. (d) Considering that the results center around an algorithm, I recommend that the author includes the algorithm as supplementary material. This would enable readers to replicate the obtained results. Additionally, it would enhance the citation of the work in the long term.
Author response: Thank you very much for your constructive comment.
Author action: To carefully address your comment, we have made modifications supplemented the relevant description in Section5.
“In Figure10, the x axis represents a sequence of tasks that request or release resources. And the y axis represents task scheduling flexibility of four algorithms mentioned in the paper.”
Reviewer #2, Concern # 7: On line 31 - "Together with the application of DPR technology,further requirements for task segmentation and reorganisation are proposed.". Make the phrase clearer. I believe the word 'needed' would be more appropriate. Sugestion: "In addition to employing DPR technology, there is a need for further requirements regarding task segmentation and reorganization"
Author response: Thank you very much for your constructive comment. We have modified this sentence according to your suggestion.
Author action: To carefully address your comment, we have made modifications to the relevant content on line 31.
“In addition to employing DPR technology, there is a need for further requirements regarding task segmentation and reorganization.”
Reviewer #2, Concern # 8: On line 33 - "The original FPGA resource management algorithms used board-level granularity for management." I believe that in this context, using the word "utilize" in the present is more appropriate than using "used" in the past.
Author response: Thank you very much for your constructive comment. We have modified this sentence according to your suggestion.
Author action: To carefully address your comment, we have made modifications to the relevant content on line 33.
“The original FPGA resource management algorithms utilize board-level granularity for management.”
Reviewer #2, Concern # 9: "the device is an empty device, and during analogue allocation it is a non-empty device." should be replaced by "the device is considered empty, and during analog allocation, it becomes non-empty.
Author response: Thank you very much for your constructive comment. We have modified this sentence according to your suggestion.
Author action: To carefully address your comment, we have made modifications to the relevant content on line 362.
“the device is considered empty, and during simulating allocation of resources, it becomes non-empty.”
Reviewer 3 Report
Comments and Suggestions for Authors
This paper proposes a geometric layout-based resource management optimization method on Multi-FPGA systems, which improves the resource flexibility and the resource utilization, and overall the resource management performance.
The paper is well organized and technically correct. The experimental part is well explained, evaluation metrics are clearly defined, and the obtained performance is compared to state-of-the-art. The literature survey is comprehensive and references are well chosen and relevant for the approached subject.
Please revise the following issues:
1) On line 139: "Pabc < Pa + Pb + Pc" ; the combined allocation is denoted Pabc, while in equation (3) and previously in text is denoted Pa+b+c; please revise and use the same notation for consistency.
2) The term "min heap" on line 156 is not explained; what does it mean?
On line 218, the term "blank resources" is not clear, please explain.
In sub-section 3.2.3, the terms "heap sequence", "heap structure", "in-heap", "out-heap" etc. are used, but they are not defined or explained; please explain them briefly for the general reader.
3) On line 167: "We define the UnAcceptance Degree (UAD) as a reasonable degree of the combination" - since it is a "reasonable degree", it should be more logical to be called "acceptance degree" (this is a suggestion)
Also on line 283: "The degree of non-acceptance" - could it be renamed "degree of acceptance" ? (just a suggestion, if possible)
4) The relation after line 149 is not numbered; number it as equation (4) and then re-number all the following equations in text.
Equation (4), which will become (5) after renumbering, should be placed where it is mentioned, at line 219, not included in section 3.3.
5) Define the variables occurring in equation (6) .
Equation (7) should be explained; what represent the parameters α and ρ and what values they may take?
Check the equation (9) if it is correct (the sum at the denominator has no argument etc.); explain the formula and its variables in more detail.
6) On line 254: "The resource request is graded by taking the base 4 logarithm of the total resource" - please explain why the base 4 logarithm is needed
On line 271: "Sort each subtask rectangle by release time" - please define the term "release time"
On line 334: explain the meaning of "analogue allocation"
7) Verify the title of Section 4 "Two Resource Management Algorithm ... " - is it correct? It is not clear what are the two algorithms.
In subsection 4.1, the acronym CLB is not explained.
On line 297, the acronym DAG ("DAG graphs") is not explained.
In text, the start and end times are denoted by small letters (sa,ea), (sb,eb) and (sc,ec), while in Figure 3 (a) start times are Sa, Sb, Sc ; please revise
8) The list of "algorithm resource requirements" given in Table 1 should be briefly explained, otherwise it is useless for the reader. Briefly compare the algorithms and explain what represent the "Slices" and "BRAM" values.
9) GRAMMAR: Although grammar is overall good, some expressions and terms are not very clear; make the following corrections suggested as follows:
line 20: more compute-intensive computing units -> more computationally-intensive (computing) units
line 23: "allows the runtime to dynamically configure tasks" - not clear, reformulate
line 29: which is a "black box" and invisible to the user -> which is a "black box" invisible to the user
line 41: "with a resource granularity of resource rectangles" - not clear, reformulate
line 60: Two resource management algorithm -> Two resource management algorithms
line 77: FPGA resource management was managed (?) -> FPGA resource management was achieved
line 78: FPGAs were grouped into a group (?) -> FPGAs were grouped / FPGAs were assembled into a group
line 84: "gradually tends to parts of the resources on-chip" - not clear, reformulate
line 89: the literature [15] -> the paper [15]
line 113: "is abstracted as a rectangle combination problem by abstracting the tasks" - reformulate
line 130: "The parameter P is used to represent the product of the subtask and the product of the start-end time interval" (?) - very unclear, reformulate (maybe "product of the subtask and start-end time interval")
line 133: resources occupied of each task is Pa, Pb and Pc -> resources occupied by each task are Pa, Pb and Pc
line 139: meets Pabc < Pa + Pb + Pc -> satisfy the condition Pabc < Pa + Pb + Pc
line 149: be expressed as formulation(??)
line 155: the acceptable combinations from these combinations -> the acceptable combinations / the acceptable combinations from these ones
line 161: to the length of the longitudinal side -> to the longitudinal side length
line 164: "Due to the dynamic reconfigurable regions of different FPGA chips have different shapes" (?) - reformulate
line 166: "and similar shapes and resource shapes" (?)
line 195: optimal solution. And the -> optimal solution, and the (don't begin a sentence with "And")
line 201: Tasks -> tasks
line 207: "largest number of sub-barrels" (?)
line 210: "child rectangles" (?) - explain the term or avoid using it
line 211: width increases less than -> width increase is less than
line 214: "it can be ensured that the increased space can place a bus" - reformulate
line 222: "Each combination will reduce the possible combination" (?) - unclear, reformulate
line 242: "longitudinal spacing of one of the rectangles is greater than the height of one of the rectangles" - unclear, reformulate
line 243: The formula is expressed as: -> This condition is expressed as:
line 271: subtaskrectangle -> subtask rectangle
line 277: "generated by task placement to place other tasks" (?)
line 308: We sets -> We set
line 321: N represents -> Mi represents
line 323: "the rationality of the task M deployed in the resource area R" - reformulate more clearly
caption of Figure 9: "maximum blank area under area under ... " (?) - revise
line 334: "the device is an empty device" (?)
line 436: In Proceedings of the Proceedings of -> In Proceedings of
10) REFERENCES:
Check that references are written in uniform style, according to the available template.
Some paper titles are in simple case (only first word beginning with a majuscule), others in title case (all words beginning with a majuscule); please check and correct.
Make the following corrections in [16]: 2d -> 2D ; fpgas -> FPGAs
Comments on the Quality of English LanguageAlthough grammar is overall good, some corrections should be made in the manuscript, as suggested in comments to authors.
Author Response
============== Authors’ responses to the reviewer #3’ comments ==============
To Reviewer 3:
The authors appreciate very much for your constructive comments on our manuscript, “GLRM: Geometric Layout-based Resource Management Method on Multi-FPGA Systems” (Manuscript ID Electronics-2959858).
We have revised our manuscript according to your comments. The revision details are summarized as follows.
--------------------------------------------------------
Reviewer #3, Concern #1: On line 139: "Pabc < Pa + Pb + Pc" ; the combined allocation is denoted Pabc, while in equation (3) and previously in text is denoted Pa+b+c; please revise and use the same notation for consistency.
Author response: Thank you for your helpful comment. We checked the relevant notation and then made the revision.
Author action: To carefully address your comment, we have modified the corresponding notation on line 147.
If the weighted resource occupancy of the resource allocation and the combined allocation meets Pa+b+c < Pa + Pb + Pc, the number of tasks accommodated by the system at the same time can be increased by the task combination, and the total resource cost of the task is reduced.
Reviewer #3, Concern #2: The term "min heap" on line 156 is not explained; what does it mean? On line 218, the term "blank resources" is not clear, please explain. In sub-section 3.2.3, the terms "heap sequence", "heap structure", "in-heap", "out-heap" etc. are used, but they are not defined or explained; please explain them briefly for the general reader.
Author response: Thank you for your constructive comment. We will provide a detailed explanation of these terms. A heap is a data structure that approximates a complete binary tree and relies on parent node key values being less than or greater than child node key values. The properties of the Min Heap satisfy the sorting required for rectangular combinatorial selection, which does not need to maintain a strict back-and-forth relationship, but only needs to obtain the key value that has the most values. The term “blank resources” represent the unused resources. A heap sequence is a linearized representation of an array that satisfies the properties of a heap. Heap structure is a special type of data structure that can be viewed as a complete binary tree and can be implemented and stored using arrays. In-heap: refers to the operation of adding a new element to the heap. Out-heap: refers to the operation of removing elements from the heap, usually the top element.
Author action: To carefully address your comment, we have supplemented these terms in this paper.
Reviewer #3, Concern #3: On line 167: "We define the UnAcceptance Degree (UAD) as a reasonable degree of the combination" - since it is a "reasonable degree", it should be more logical to be called "acceptance degree" (this is a suggestion). Also on line 283: "The degree of non-acceptance" - could it be renamed "degree of acceptance" ? (just a suggestion, if possible)
Author response: Thank you for your constructive comment. We will take your suggestion and change the UnAcceptance Degree (UAD) to Acceptance Degree (AD).
Author action: To carefully address your comment, We have made modifications to the corresponding wording like on line 175.
“We define the Acceptance Degree (AD) as a reasonable degree of the combination that can be expressed as formulation(6)”
Reviewer #3, Concern #4: The relation after line 149 is not numbered; number it as equation (4) and then re-number all the following equations in text. Equation (4), which will become (5) after renumbering, should be placed where it is mentioned, at line 219, not included in section 3.3.
Author response: Thank you for your helpful comment. We have checked the formula numbering throughout the entire text and renumbered it in order. In addition, we have also adjusted the position of some formulas.
Author action: To carefully address your comment, we have made corrections to the issues mentioned above.
Reviewer #3, Concern #5: Define the variables occurring in equation (6). Equation (7) should be explained; what represent the parameters α and β and what values they may take? Check the equation (9) if it is correct (the sum at the denominator has no argument etc.); explain the formula and its variables in more detail.
Author response: Thank you for your helpful comment. In equation (6), B1 represents the bus bandwidth, B2 represents the on-chip resource bandwidth. In equation (7), α represents the weight of resource fragments in calculating acceptability, β represents the weight of the difference between RWH and SRWH, the range of values is all from 0 to 1. Due to our negligence, there was a writing error in Formula 9. The correct way to write it should be:
Author action: To carefully address your comment, we have supplemented and modified the relevant formula content in this paper.
“where α denotes the weight of the resource fragment when calculating the acceptability, and β denotes the weight of the difference value between RWH and SRWH, and the range of α and β is all from 0 to 1.”
Reviewer #3, Concern #6: On line 254: "The resource request is graded by taking the base 4 logarithm of the total resource" - please explain why the base 4 logarithm is needed. On line 271: "Sort each subtask rectangle by release time" - please define the term "release time" On line 334: explain the meaning of "analogue allocation".
Author response: Thank you for your constructive comment. By taking the logarithm of the total resources with a base of 4, resource requests are graded, which is more in line with the quadtree resource management approach and also convenient for computation. Release time refers to the time when the task requests resources. Analogue allocation is a method of simulating experiments or calculations used to test and optimize the efficiency and performance of the actual task allocation process.
Author action: To carefully address your comment, we have supplemented the relevant content in Section5.3 on lines 359-362.
“We simulate the allocation of resources according to the task sequence which is the sequence generated by the full alignment of the 11 tasks and then generate resource layouts according to the resource management algorithms before and after improvement respectively.”
Reviewer #3, Concern #7: Verify the title of Section 4 "Two Resource Management Algorithm ... " - is it correct? It is not clear what are the two algorithms. In subsection 4.1, the acronym CLB is not explained. On line 297, the acronym DAG ("DAG graphs") is not explained. In text, the start and end times are denoted by small letters (sa,ea), (sb,eb) and (sc,ec), while in Figure 3 (a) start times are Sa, Sb, Sc ; please revise.
Author response: Thank you for your constructive comment. In order to make the meaning of the Section 4’s title more clear, we have modified the title to "The Quadtree and Central Bus Resource Management Algorithm Optimized by TCA". The full name of CLB is configurable logic blocks. The full name of a DAG graph is a directed acyclic graph. We have modified the parameter form in Figure 3 to match the parameter form in the text.
Author action: To carefully address your comment, we have supplemented the relevant content in Section 3.1 on line 141.
“the start and end times of each subtask in the system are (Sa, Ea), (Sb, Eb) and (Sc, Ec) respectively.”
Reviewer #3, Concern #8: The list of "algorithm resource requirements" given in Table 1 should be briefly explained, otherwise it is useless for the reader. Briefly compare the algorithms and explain what represent the "Slices" and "BRAM" values.
Author response: Thank you for your constructive comment. In Table 1, common algorithms for target recognition include Debayer, Rectifier, Stereo match, Disparity, Flex-SURF and Motor Control. And common algorithms for image processing include FPN correction, Dark field corr. , FFT, Bad pixel/spike CCSDS 122, Binning, Hough Transform and Median Filter. “Slices” represents the unit of measure for logic cells in FPGAs. The full name of BRAM is block RAM. “BRAM” is the predefined hardware resource in the FPGA that is dedicated to storage.
Author action: To carefully address your comment, we have supplemented the relevant content on lines 330-335.
“And common algorithms for image processing include FPN correction, Dark field corr. , FFT, Bad pixel/spike CCSDS 122, Binning, Hough Transform and Median Filter. Slice represents the unit of measure for logic cells in FPGAs. The full name of BRAM is block RAM. BRAM is the predefined hardware resource in the FPGA that is dedicated to storage.”
Reviewer #3, Concern #9: GRAMMAR: Although grammar is overall good, some expressions and terms are not very clear; make the following corrections suggested as follows:
line 20: more compute-intensive computing units -> more computationally-intensive (computing) units
......
line 436: In Proceedings of the Proceedings of -> In Proceedings of
Author response: Thank you for your constructive comment. In order to make expressions and terms clear, we have checked the expressions and terms in the paper and modified corrections.
Author action: To carefully address your comment, we have made modifications to the relevant expressions and terms in the paper.
Reviewer #3, Concern #10: REFERENCES:Check that references are written in uniform style, according to the available template. Some paper titles are in simple case (only first word beginning with a majuscule), others in title case (all words beginning with a majuscule); please check and correct.Make the following corrections in [16]: 2d -> 2D ; fpgas -> FPGAs
Author response: Thank you for your constructive comment. We have checked the references to make them be written in uniform style and correct.
Author action: To carefully address your comment, we have made modifications to the relevant references in the paper.
“Cui, J.; Deng, Q.; He, X.; Gu, Z. An efficient algorithm for online management of 2D area of partially reconfigurable FPGAs. In Proceedings of the 2007 Design, Automation & Test in Europe Conference & Exhibition. IEEE, 2007, pp. 1–6.”
Reviewer 4 Report
Comments and Suggestions for Authors
The article proposes the design of a new geometric layout-based task combination algorithm to use the available FPGA resources (of Multi-FPGA systems) more efficiently. Also, the authors propose the improvement of two resource management algorithms and provide measurable figures for the resulting improvements.
The related work section is too short. It may be improved by adding more details and explainations on the current FPGA resource management techniques and processes.
Section 3 describes in length the Geometric Layout-based Task Combination Algorithm. The presentation uses too much textual description. It is recommended to use a common algorithmic description with pseudocode and emphasis on the steps of the algorithm.
Section 4 may use the same algorithmic description as recommended for section 3.
On line 168 there is a formula that mentions alpha and beta parameters. Their specific usage must be better explained. The same observation holds for formula 7 where alpha and rho parameters need additional explanations.
Explain the claim in line 357.
Figure 10 must have the x axis better explained.
Typo on line 149: “??”
Typo on line 308: “We sets the common…”
Only 5 out of 22 references are from the last five years.
Overall, the article is well written, but it feels too expeditive. Some sections must be extended and additional descriptions/explanations must be provided.
Comments on the Quality of English LanguageMinor editing of English language required
Author Response
============== Authors’ responses to the reviewer #4’ comments ==============
To Reviewer 4:
The authors appreciate very much for your constructive comments on our manuscript, “GLRM: Geometric Layout-based Resource Management Method on Multi-FPGA Systems” (Manuscript ID Electronics-2959858).
We have revised our manuscript according to your comments. The revision details are summarized as follows.
--------------------------------------------------------
Reviewer #4, Concern #1: The related work section is too short. It may be improved by adding more details and explainations on the current FPGA resource management techniques and processes.
Author response: Thank you very much for your constructive comment. We conducted a more extensive search on academic websites to further supplement for relevant literature on the current FPGA resource management techniques and processes. However, limited by the niche of research content, there is only a small amount of relevant research progress available for reference and comparison. For this, we have added a few new relevant literature and details to improve related work sections.
Author action: To carefully address your comment, the additions are in lines 99-106.
[21] proposed a novel 100area sharing method that leverages the exploratory capabilities of OpenCL, employing intelligent clustering and custom, task-specific partitioning and mapping to more effectively manage the resource requirements of tasks. By selecting the most appropriate distribution that best enhances the temporal compute density based on runtime workload demands, the system’s throughput is improved. Mehrabi[22] highlighted the importance of considering spatio-temporal strategies in FPGA resource management to achieve long-term target allocations and optimize FPGA usage efficiency by tracking and correcting deviations.
Reviewer #4, Concern #2: Section 3 describes in length the Geometric Layout-based Task Combination Algorithm. The presentation uses too much textual description. It is recommended to use a common algorithmic description with pseudocode and emphasis on the steps of the algorithm.
Author response: Thank you very much for your constructive comment. We have added the algorithm description with pseudocode and emphasis on the steps of the algorithm you have recommended in Section3.
Author action: To carefully address your comment, we have made modifications to the algorithm description as shown on Algorithm3.1 in Section3.
Reviewer #4, Concern #3: Section 4 may use the same algorithmic description as recommended for section 3.
Author response: Thank you very much for your constructive comment. We have added the algorithm description with pseudocode and emphasis on the steps of the algorithm you have recommended in Section4.
Author action: To carefully address your comment, we have made modifications to the algorithm description as shown on Algorithm4.1 in Section4.
Reviewer #4, Concern #4: On line 168 there is a formula that mentions alpha and beta parameters. Their specific usage must be better explained. The same observation holds for formula 7 where alpha and rho parameters need additional explanations.
Author response: Thank you very much for your constructive comment. We have checked and corrected the mathematical expressions in the paper. α represents the weight of resource fragments in calculating acceptability, β represents the weight of the difference between RWH and SRWH, the range of values is all from 0 to 1. And α , β in formula7 have the same explanations as described above.
Author action: To carefully address your comment, we have checked and added the relevant explanation of parameters in all formula.
“where ADi is the unacceptance degree of rectangle i, α denotes the weight of the resource fragment when calculating the acceptability, β denotes the weight of the difference value between RWH and SRWH.”
Reviewer #4, Concern #5: Figure 10 must have the x axis better explained.
Author response: Thank you very much for your constructive comment. We have checked and added the axis of figures in the paper. In Figure10, the x axis represents a sequence of tasks that request or release resources. And the y axis represents task scheduling flexibility of four algorithms mentioned in the paper.
Author action: To carefully address your comment, we have checked and added the relevant explanation of axis in figures.
“In Figure10, the x axis represents a sequence of tasks that request or release resources. And the y axis represents task scheduling flexibility of four algorithms mentioned in the paper.”
Reviewer #4, Concern #6: Typo on line 149: “??”, Typo on line 308: “We sets the common…”
Author response: Thank you for your constructive comment. In order to make expressions and terms clear, we have checked the expressions and terms in the paper and modified corrections.
Author action: To carefully address your comment, we have made modifications to the relevant expressions and terms in the paper.
“We set the common algorithms for target recognition and image processing as task topology generated by TGFF”
Reviewer #4, Concern #7: Only 5 out of 22 references are from the last five years.
Author response: Thank you for your constructive comment. Noting this, we reviewed other academic papers to further refer more current researches, but due to the paucity of research in this area of study in recent years, only a limited number of relevant articles are available. For this, we have added a few new relevant literature and details to this paper.
Author action: To carefully address your comment, we have we have added a few current relevant literature and details to this paper.
“Wang, Y.; Liao, Y.; Yang, J.; Wang, H.; Zhao, Y.; Zhang, C.; Xiao, B.; Xu, F.; Gao, Y.; Xu, M.; et al. An FPGA-based online reconfigurable CNN edge computing device for object detection. Microelectronics Journal 2023, 137, 105805.”
“Mehrabi, A.; Sorin, D.J.; Lee, B.C. Spatiotemporal Strategies for Long-Term FPGA Resource Management. In Proceedings of the 2022 IEEE International Symposium on Performance Analysis of Systems and Software (ISPASS), 2022, pp. 198–209. https://doi.org/10.1109/ISPASS55109.2022.00026.”
Round 2
Reviewer 1 Report
Comments and Suggestions for Authors
I am seeing your explanations of my comments and the algorithms added to the paper to improve the reading of the document. It will help the readers to understand your work in a better way. I have no more comments.
Reviewer 2 Report
Comments and Suggestions for Authors
The authors significantly improved the manuscript by providing better clarification of the work's contribution in the introduction and making the other requested adjustments throughout the text. I believe the article is now at the desired level for Electronics
Comments on the Quality of English LanguageNo comments
Reviewer 4 Report
Comments and Suggestions for Authors
The article proposes the design of a new geometric layout-based task combination algorithm to use the available FPGA resources (of Multi-FPGA systems) more efficiently.
The authors properly addressed all the comments.
The algorithmic description that was requested is now part of the article.
The introduction offers a slightly widened context.
The article quality has been improved and can be accepted in present form.